# Systematic Review with Meta-Analyses: Diagnostic Accuracy of FibroMeter Tests in Patients with Non-Alcoholic Fatty Liver Disease

**DOI:** 10.3390/jcm10132910

**Published:** 2021-06-29

**Authors:** Anne-Marieke Van Dijk, Yasaman Vali, Anne Linde Mak, Jenny Lee, Maarten E. Tushuizen, Mohammad Hadi Zafarmand, Quentin M. Anstee, M. Julia Brosnan, Max Nieuwdorp, Patrick M. Bossuyt, Adriaan G. Holleboom

**Affiliations:** 1Department of Internal and Vascular Medicine, Amsterdam University Medical Centres, Location AMC, 1105AZ Amsterdam, The Netherlands; a.l.mak@amsterdamumc.nl (A.L.M.); m.nieuwdorp@amsterdamumc.nl (M.N.); a.g.holleboom@amsterdamumc.nl (A.G.H.); 2Department of Epidemiology and Data Science, Amsterdam University Medical Centres, Location AMC, 1105AZ Amsterdam, The Netherlands; y.vali@amsterdamumc.nl (Y.V.); j.a.lee@amsterdamumc.nl (J.L.); m.h.zafarmand@amsterdamumc.nl (M.H.Z.); p.m.bossuyt@amsterdamumc.nl (P.M.B.); 3Department of Gastroenterology and Hepatology, Leiden University Medical Centre, 2333ZA Leiden, The Netherlands; M.E.Tushuizen@lumc.nl; 4Translational & Clinical Research Institute, Faculty of Medical Sciences, Newcastle University, Newcastle upon Tyne NE1 7RU, UK; quentin.anstee@newcastle.ac.uk; 5Internal Medicine Research Unit, Pfizer Inc., Cambridge, MA 02139, USA; julia.brosnan@pfizer.com

**Keywords:** non-invasive test, biomarker, fatty liver, liver fibrosis, non-alcoholic steatohepatitis

## Abstract

Early detection of liver fibrosis is crucial to select the correct care path for patients with non-alcoholic fatty liver disease (NAFLD). Here, we systematically review the evidence on the performance of FibroMeter versions in detecting different levels of fibrosis in patients with NAFLD. We searched four databases (Medline, Embase, the Cochrane library, and Web of Science) to find studies that included adults with NAFLD and biopsy-confirmed fibrosis (F1 to F4), compared with any version of FibroMeter. Two independent researchers screened the references, collected the data, and assessed the methodological quality of the included studies. We used a bivariate logit-normal random effects model to produce meta-analyses. From 273 references, 12 studies were eligible for inclusion, encompassing data from 3425 patients. Meta-analyses of the accuracy in detecting advanced fibrosis (F ≥ 3) were conducted for FibroMeter Virus second generation (V2G), NAFLD, and vibration controlled transient elaFS3stography (VCTE). FibroMeter VCTE showed the best diagnostic accuracy in detecting advanced fibrosis (sensitivity: 83.5% (95%CI 0.58–0.94); specificity: 91.1% (95%CI 0.89–0.93)), followed by FibroMeter V2G (sensitivity: 83.1% (95%CI 0.73–0.90); specificity: 84.4% (95%CI 0.62–0.95)) and FibroMeter NAFLD (sensitivity: 71.7% (95%CI 0.63–0.79); specificity: 82.8% (95%CI 0.71–0.91)). No statistically significant differences were found between the different FibroMeter versions. FibroMeter tests showed acceptable sensitivity and specificity in detecting advanced fibrosis in patients with NAFLD, but an urge to conduct head-to-head comparison studies in patients with NAFLD of the different FibroMeter tests remains.

## 1. Introduction

Non-alcoholic fatty liver disease (NAFLD) is one of the main causes of chronic liver disease, with a global prevalence of 25% [1]. It is highly associated with insulin resistant states, including the metabolic syndrome, type 2 diabetes mellitus (T2DM), and obesity. These conditions are increasing, driving the prevalence of progressive stages of NAFLD. The disease spectrum of NAFLD ranges from simple steatosis to non-alcoholic steatohepatitis (NASH), characterized by hepatocyte ballooning and lobular inflammation, followed by stages of fibrosis and even cirrhosis [2,3]. Liver fibrosis strongly correlates with increased overall- and liver-related mortality and liver-related complications including hepatocellular carcinoma (HCC) and the need for liver transplantation [4,5]. Therefore, international guidelines recommend the early identification of NAFLD and associated liver fibrosis, especially the advanced stages of bridging fibrosis (F3) and cirrhosis (F4) [6,7,8].

Currently, liver biopsy is considered the gold-standard for staging liver fibrosis in patients with chronic liver diseases including NAFLD. However, it is an invasive and relatively costly procedure, with risks of postprocedural bleeding, sampling error and interobserver variability [6,9]. Guidelines recommend non-invasive tests to distinguish patients at low risk of advanced fibrosis from those at high risk [6,7,8]. The rigorous validation of such non-invasive proxies of liver fibrosis will enhance their implementation in care paths for NAFLD.

FibroMeter is a group of liver fibrosis proxies composed of different sets of biometric and blood parameters, aimed to estimate fibrosis in different causes of chronic liver disease [10]. Currently, different variants of FibroMeter are being used in clinical settings to cover main liver disease etiologies, including viral hepatitis, NAFLD and alcoholic liver disease.

FibroMeter Virus second generation (V2G) was initially developed for the diagnosis of significant fibrosis in patients with hepatitis C and includes the following parameters: age, sex, aspartate transaminase (AST), urea, platelets, prothrombin time, hyaluronate, and alpha2-macroglobulin [11], of which the last two parameters are direct markers of liver fibrosis [12]. In FibroMeter Virus third generation (V3G), hyaluronate was replaced by gamma-glutamyl transferase (GGT), for which diagnostic processes are more generally available. In recent years, FibroMeter V2G and V3G have also been validated in patients with NAFLD [12,13]. Recently, to increase performance of this blood test, an algorithm that combines FibroMeter V3G with the liver stiffness measured by vibration controlled transient elastography (VCTE) was introduced, called FibroMeter VCTE [14]. VCTE measures shear wave velocity and determines tissue stiffness by ultrasound [15]. FibroMeter NAFLD is another version of FibroMeter that was developed especially for NAFLD. It includes more commonly used parameters such as age, body weight, AST, alanine transaminase (ALT), platelets, glucose, and ferritin [16].

A number of studies evaluated the accuracy of different versions of FibroMeter in patients with NAFLD. Some evidence suggests that the performance of FibroMeter NAFLD is comparable to four commonly used non-invasive fibrosis scores: (1) the NAFLD fibrosis score (NFS); (2) the AST- platelet ratio (APRI); (3) the body mass index (BMI), AST/ALT ratio and diabetes (BARD) score; and (4) the Fibrosis-4 (FIB-4) score [17,18]. On the other hand, there is also evidence showing significantly better performance in diagnosing NAFLD fibrosis of FibroMeter V2G compared to other non-invasive tests, including FibroMeter NAFLD [13]. This is surprising given that FibroMeter V2G was originally designed to diagnose hepatitis C, while FibroMeter NAFLD was designed specifically for the purpose of diagnosing NALFD fibrosis.

Because of these conflicting results and to explore other likely sources of variability in the accuracy of reported tests, we aimed to perform a systematic review and meta-analysis on the diagnostic accuracy of multiple FibroMeter versions for fibrosis staging among patients with NAFLD.

## 2. Materials and Methods

The present study has been conducted as part of a large multicentre project named LITMUS (Liver Investigation: Testing Marker Utility in Steatohepatitis), which is funded by the European Union IMI2 scheme, aiming to develop, validate, and qualify a defined set of biomarkers that enable detection of the NAFLD spectrum.

The protocol of the full systematic review is available in PROSPERO: CRD42018106821. This study is conducted according to the registered protocol. This study report was prepared using the PRISMA-DTA statement, see Appendix A [19].

### 2.1. Search Strategy and Screening

We conducted a search on Medline (via OVID), Embase (via OVID), the Cochrane library, and Web of Science to find potentially eligible studies, in December 2018. This search was initially conducted for different markers, containing words in the title/abstract or text words across the record and the medical subject heading (MeSH). The search strategy was developed in collaboration with an experienced medical librarian, whereby input from the study investigators was used. The details of the search strategy are included in the Appendix A [20,21]. The search was updated in October 2020 specifically for FibroMeter. Our search was limited to human subjects, however we applied no further restrictions based on either year or language.

Search results of all databases were combined, and duplicates were removed using Endnote software. Two independent authors screened the titles and abstracts of articles identified in the search (A.L.M., A.-M.v.D.) In the full text screening phase, all remaining articles were thoroughly examined for eligibility by the same reviewers independently, including reference lists. The reference list was screened to find possible relevant articles not detected by the search. In the case of disagreement about eligibility, studies were discussed between the two reviewers and, if needed, with the third reviewer (Y.V.) to reach agreement on inclusion or exclusion. The title and abstract screening phase was conducted using Rayyan QCRI (https://rayyan.qcri.org (accessed on 14 June 2021)). Next, reference lists of related systematic reviews and included studies were manually searched to identify additional studies. In addition, we contacted partners within the LITMUS consortium for any study that may have been missed.

### 2.2. Inclusion and Exclusion Criteria

Eligible for inclusion in this systematic review were diagnostic accuracy studies reported in peer-reviewed journals or conference abstracts in adult patients (≥18 years of age) with NAFLD, that evaluated any version of FibroMeter against liver biopsy as the clinical reference standard, with a maximum time window of six months between blood sampling and biopsy collection. Patient results only reported or mentioned in letters or commentaries were excluded. Diagnostic accuracy was expressed in terms of the area under the receiver operating characteristic curve (AUC) with its 95% confidence interval (95% CI).

The target conditions of interest were NASH and fibrosis (F1–F4). All studies were included regardless of the NASH or fibrosis scoring systems used. Studies that had included patients with mixed aetiologies (e.g., viral hepatitis) were only included if outcomes were separately reported for patients with NAFLD. In case of overlapping patient groups, the study with the largest number of patients was selected. For the meta-analysis, we excluded studies with insufficient information to reconstruct a 2 × 2 contingency table after contacting the study authors. The source of funding and conflict of interest were also collected in each study.

### 2.3. Data Extraction and Quality Assessment

The following data were independently extracted from the report for each included study: study characteristics, clinical characteristics, index test and liver biopsy features, and data necessary to construct a 2 × 2 contingency table (true positives, true negatives, false positives and false negatives) to assess the performance of the index test. All data were extracted by one author (A.-M.v.D.) and verified by another (A.L.M.) If studies used METAVIR as histological scoring system for staging fibrosis, these scores were converted to their NASH CRN equivalent using the conversion suggested by Boursier and colleagues (see Appendix A) [22]. We contacted authors of included studies in case more information was required to construct 2 × 2 contingency tables (see Acknowledgments).

The Quality Assessment of Diagnostic Accuracy Studies tool (QUADAS-2) was used to assess methodological quality of all included studies [23]. Two reviewers independently evaluated the risk of bias and applicability concerns of each included study (A.L.M., A.-M.v.D.) A judgment of ‘low’, ‘high’ or ‘unclear risk’ was assigned to each of the QUADAS-2 domains, resulting in an overall risk of bias per domain and applicability judgment for each study.

### 2.4. Statistical Analyses

The sensitivity and specificity with 95% CI were graphically summarized as forest plots for each reported cut-off using RevMan software [24] Accuracy data were extracted or reconstructed for reported cut-offs according to the original studies.

Included studies were classified into groups for meta-analysis, based on the pre-defined target conditions (significant fibrosis (F ≥ 2) and advanced fibrosis (F ≥ 3)) and the various FibroMeter versions: V2G, V3G, NAFLD and VCTE. We applied a bivariate logit-normal random-effects model to compute summary estimates of sensitivity, specificity and AUC. With these, we calculated predictive values. We defined the minimal acceptable performance levels of sensitivity, specificity and AUC estimates for any FibroMeter version as ≥ 0.80 [21]. Summary receiver operating characteristic (SROC) curves were constructed to represent the overall diagnostic accuracy of the index test. The statistical imprecision about the location of the mean was expressed by calculating confidence intervals around the summary point. Furthermore, we computed the 95% prediction region around the summary point, to indicate where we expect the true sensitivity and specificity of similar future studies to fall. We assessed the heterogeneity based on visual assessment of forest plots and ROC curves.

Publication bias was not formally evaluated by statistical tests or in a funnel plot, because no accepted statistical test can reliably discriminate publication bias from other sources of bias in diagnostic meta-analyses [25].

### 2.5. Additional Analyses

In an outlier analysis, we reran the meta-analyses after excluding one study with very low sensitivity.

### 2.6. Meta-Regression Analysis

We compared the performance of the different versions of FibroMeter in a meta-regression analysis with a bivariate logit-normal random-effects model. Including studies that provided a 2 × 2 table for one or two FibroMeter tests, we estimated mean logit sensitivity and specificity, estimates of the differences in the mean, and the respective variances and covariance. *p* values < 0.05 were considered to indicate statistically significant differences. R for Windows (Version 3.6.0; R Foundation for Statistical Computing, Vienna, Austria) was used for all analyses.

## 3. Results

### 3.1. Search Results

Our initial search for accuracy studies of different NAFLD biomarkers listed by LITMUS identified 9066 studies. After removal of duplicates, 6220 titles remained for screening, yielding 778 eligible abstracts. Subsequent screening of these abstracts resulted in 265 studies suitable for full-text screening. A total of 15 studies were identified through this initial search for different FibroMeter versions. Our search update in October 2020 resulted in six additionally eligible studies. Full texts of 21 studies were screened and, in total, 12 studies evaluating the accuracy of one or more FibroMeter versions were included in this systematic review: ten full reports [13,17,18,26,27,28,29,30,31,32] and two abstracts [33,34]. Eleven studies were eligible for inclusion in our meta-analyses. See the flow diagram (Figure 1) for a complete representation of the applied search strategy and reasons for exclusion.

### 3.2. Study Characteristics

A summary of the characteristics of the included studies can be found in Table 1. All studies were conducted in tertiary care centres. One abstract did not report this information [33]. Eight studies included the FibroMeter especially designed for NAFLD (FibroMeter NAFLD) [13,17,18,27,29,30,32,34], four studies used the FibroMeter VCTE [26,28,29,33], three studies the FibroMeter V2G [13,27,32], and one study the FibroMeter V3G [31]. Four studies assessed more than one FibroMeter test [13,26,29,31]. Two studies had a possible conflict of interest, because of consulting activities for Echosens, a company that has a license for FibroMeter [13,26].

### 3.3. Target Population

In total, data from 3425 patients in 12 studies were included in this systematic review. The mean age of the participants varied from 37 to 65 years. In all included studies, except one, more than half of the patients were males [30]. Most patients had a high mean BMI (>25) in all studies. The highest average was 35.8 kg/m^2^ [30], while the lowest mean BMIs were reported by two studies from China: 25.9 and 26.8 kg/m^2^ [29,32].

### 3.4. Target Conditions

Six studies investigated significant fibrosis (F ≥ 2), while accuracy in detecting advanced fibrosis (F ≥ 3) was reported by 11 studies. Three studies evaluated accuracy of detecting advanced fibrosis (F ≥ 3) with FibroMeter V2G [13,26,31], one with FibroMeter V3G [31], seven with FibroMeter NAFLD [13,17,18,27,29,30,32] and four with FibroMeter VCTE [26,28,29,33]. This allowed us to perform three meta-analyses for the advanced fibrosis target condition among three different versions of FibroMeter (V2G, NAFLD, VCTE) (see overview Appendix A).

### 3.5. Characteristics of Liver Biopsy

Not all studies provided detailed information about the liver biopsy. Seven studies reported time interval between biopsy and blood sampling, with a range from one week up to a maximum of three months [13,26,27,29,30,31,32] Eight studies indicated that the histological samples were assessed with no knowledge of the patients’ clinical data [13,15,26,27,30,31,32,34]. Three studies reported liver biopsy evaluation by one pathologist specialized in hepatology [13,26,30] and four studies reported histological assessment by two pathologists [29,31,32,33]. See overview in Appendix A. Ten studies used the NASH CRN fibrosis scoring system. In one abstract, the fibrosis classification was not specified [33]. One study used the METAVIR criteria for staging fibrosis. The histological scores from this study were converted to their NASH CRN equivalent before the meta-analysis (see Appendix A) [27].

### 3.6. Methodological Quality Assessment

The methodological quality of the ten full text articles is summarized in Appendix A. Five studies were assessed to have an unclear risk of bias and four studies to have a high risk of bias in the ‘Index test’ domain. The index test was scored as ‘high’ or ‘unclear risk’ most often, which can be attributed to the absence of a suggested cut-off for FibroMeter. We had no applicability concerns in the patient selection, index test, or reference standard domains for any of the studies, because of the clear use of selection criteria for patient inclusion and proper use of the index test.

### 3.7. Accuracy of FibroMeter in Detecting Significant Fibrosis (F ≥ 2)

Significant fibrosis was reported by six studies using three different versions of FibroMeter (FM-V2G, FM-NAFLD, and FM-VCTE) [13,17,26,28,29,31]. Although we were able to extract contingency tables from four studies, the insufficient number of studies reported accuracy data for each FibroMeter version impeded conducting a meta-analysis for this target condition. The prevalence of significant fibrosis ranged from 32% to 68%. See Table 2 for area under the receiver operating characteristic curve (AUC) of different FibroMeter versions in detecting significant fibrosis reported by six studies [13,17,26,28,29,31].

#### 3.7.1. FibroMeter V2G

Four studies evaluated performance of FibroMeter V2G in detecting significant fibrosis. Cut-off values spanned from 0.31 to 0.77 [26,31], while AUC ranged from 0.75 to 0.86 [13,26,31].

#### 3.7.2. FibroMeter NAFLD

From three studies that investigated accuracy of FibroMeter NAFLD in detecting significant fibrosis, two reported a cut-off of 0.62 [17,29] and one failed to report any cut-off value. The AUCs reported by these three studies ranged from 0.62 to 0.78 [13,17,29]. One study evaluated the performance of FibroMeter NAFLD in detecting significant fibrosis combined with NASH (fibrotic NASH) and reported an AUC of 0.72 at a cut-off of 0.62 [34].

#### 3.7.3. FibroMeter VCTE

All three studies that assessed diagnosis of significant fibrosis using FibroMeter VCTE reported a cut-off value, ranging from 0.38 to 0.69 and AUC levels ranged from 0.82 to 0.86 [26,28,29].

### 3.8. Accuracy of FibroMeter in Detecting Advanced Fibrosis (F ≥ 3)

Accuracy in detecting advanced fibrosis (F ≥ 3) was reported by 11 studies with a relatively wide range of prevalence, from 19% to 41%.

#### 3.8.1. FibroMeter V2G

Three studies were included in the meta-analysis of the diagnostic accuracy of FibroMeter V2G in detecting advanced fibrosis, including 1576 patients with NAFLD, of whom 604 had advanced fibrosis (see Appendix A). Each study reported different cut-off values for this FibroMeter version; 0.39, 0.45, and 0.77 [13,26,31]. The meta-analysis resulted in combined AUC of 0.89 and mean sensitivity and specificity of 0.83 (95%CI 0.73–0.90) and 0.84 (95%CI 0.62–0.95), respectively (Appendix A). See Figure 2 for the SROC and Appendix A for the forest plot.

#### 3.8.2. FibroMeter NAFLD

Seven studies were included in the meta-analysis of FibroMeter NAFLD in detecting advanced fibrosis, having recruited 1616 patients, of whom 514 had advanced fibrosis [13,17,18,27,29,30,32]. Cut-off values ranged from 0.31 to 0.83, where 0.82 was the most commonly used cut-off reported by the three studies [17,18,29]. Siddiqui et al. reported three different cut-off values, see the forest plot in Appendix A. In the meta-analysis, we evaluated performance at the cut-off value of 0.59 [30]. This value equals the maximal Youden index of this study. Our meta-analysis resulted in estimates of the mean sensitivity of 0.65 (95%CI 0.51–0.77) and specificity of 0.86 (95%CI 0.75–0.93), while the summary estimate of the AUC was 0.82. See Figure 3a,b and Appendix A for the details of the studies and the SROC, including additional analysis described below and showed in Figure 3b.

#### 3.8.3. FibroMeter VCTE

Four studies were included in the meta-analysis of FibroMeter VCTE, including 1546 NAFLD patients of whom 542 had advanced fibrosis, see Appendix A [26,28,29,33]. A wide and heterogeneous range of cut-off values was reported by the studies, from 0.38 to 0.80. One study reported more than one cut-off value for the target condition advanced fibrosis, however the Youden-cut-off was not mentioned. See the forest plot of the studies in the Appendix A [33]. We selected the highest cut-off of 0.67 for our meta-analysis, because this was the closest value to the other reported cut-offs. Therefore, the cut-off included in the meta-analysis ranged from 0.67 to 0.80. We calculated an AUC of 0.94, sensitivity of 0.70 (95% CI 0.33–0.92) and specificity of 0.93 (95%CI 0.88–0.96) for detecting advanced fibrosis with FibroMeter VCTE. See Figure 4a for the SROC.

**Figure 2 jcm-10-02910-f002:**
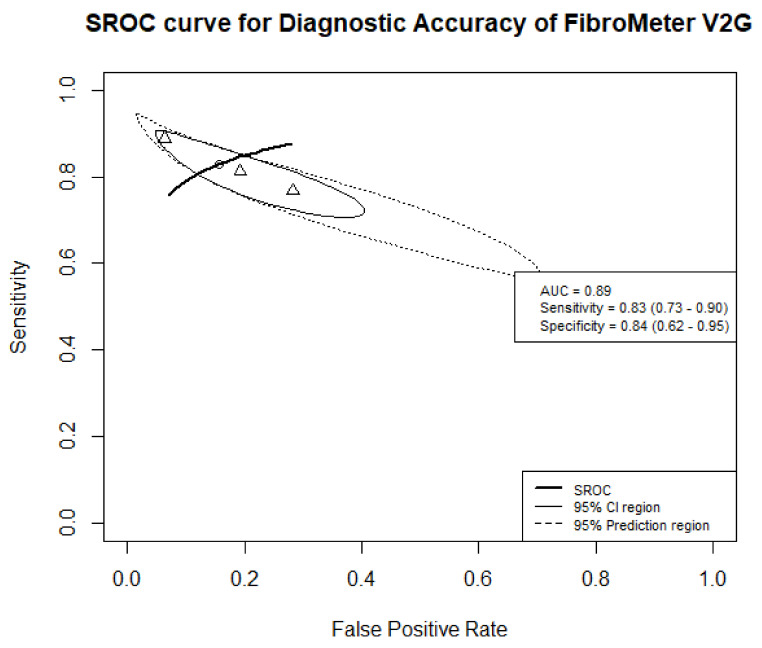
Summary receiver operating characteristic (SROC) curve of the diagnostic accuracy of FibroMeter V2G for target condition advanced fibrosis. The solid ellipse presents the 95% confidence interval region of diagnostic accuracy of FibroMeter V2G; the dotted ellipse presents the prediction region in which 95% of future diagnostic accuracy study estimates of FibroMeter V2G will fall. Triangles represent diagnostic accuracy estimates from each included study; circle presents the Youden Index threshold value.

### 3.9. Additional Analyses

We observed no statistically significant changes in the overall meta-analysis results for FibroMeter NAFLD and VCTE by removing one study with very low sensitivity estimate [30]. The sensitivity of FibroMeter NAFLD increased from 0.65 to 0.72 (95%CI 0.63–0.79), while the specificity slightly decreased from 0.86 to 0.83 (95%CI 0.71–0.91). For FibroMeter VCTE, we observed an increase in sensitivity from 0.70 to 0.84 (95%CI 0.58–0.954) and a slight decrease of specificity from 0.93 to 0.91 (95%CI 0.89–0.93) (see Figure 3 and Figure 4 and Appendix A).

**Figure 3 jcm-10-02910-f003:**
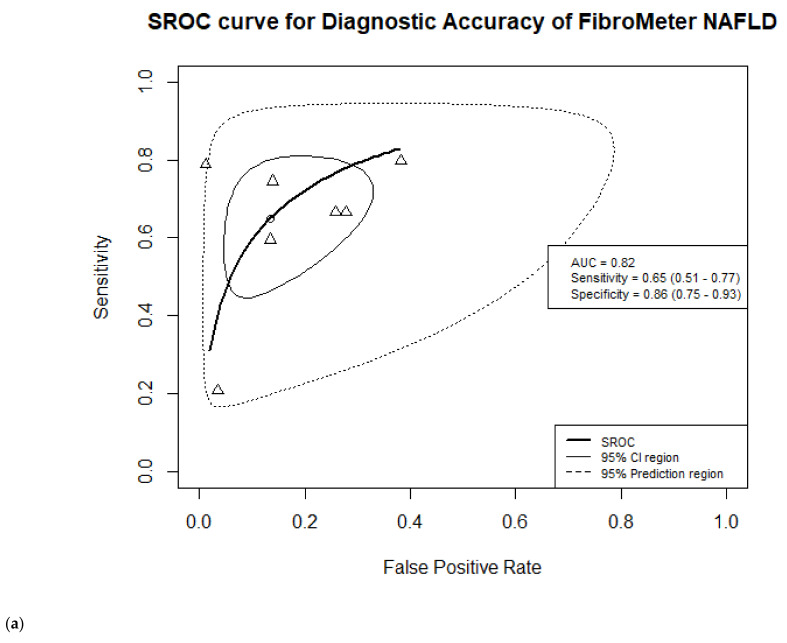
Summary receiver operating characteristic (SROC) curve of the diagnostic accuracy of FibroMeter NAFLD for target condition advanced fibrosis. The solid ellipse presents the 95% confidence interval region of diagnostic accuracy of FibroMeter NAFLD; the dotted ellipse presents the prediction region in which 95% of future diagnostic accuracy study estimates of FibroMeter NAFLD will fall. Triangles represent diagnostic accuracy estimates from each included study; circle presents the Youden Index threshold value. (**a**) Before additional sensitivity analyses; (**b**) After additional sensitivity analyses, without study of Loong et al. [29].

**Figure 4 jcm-10-02910-f004:**
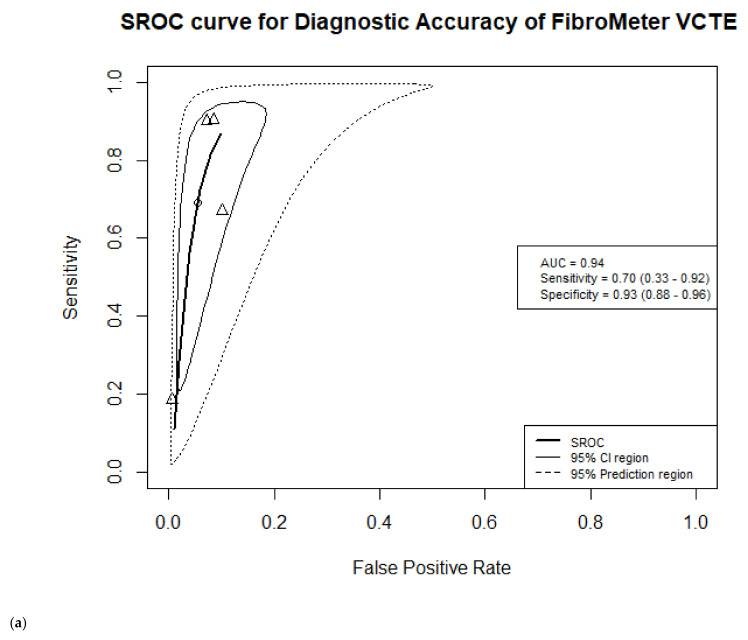
Summary receiver operating characteristic (SROC) curve of the diagnostic accuracy of FibroMeter VCTE for target condition advanced fibrosis. The solid ellipse presents the 95% confidence interval region of diagnostic accuracy of FibroMeter VCTE; the dotted ellipse presents the prediction region in which 95% of future diagnostic accuracy study estimates of FibroMeter VCTE will fall. Triangles represent diagnostic accuracy estimates from each included study; circle presents the Youden Index threshold value. (**a**) Before additional sensitivity analyses; (**b**) After additional sensitivity analyses, without study of Loong et al. [29].

## 4. Discussion

In this systematic review and accompanying meta-analyses, we summarized the available evidence on the diagnostic accuracy of the various FibroMeter versions, namely V2G, VCTE, and NAFLD, in detecting NAFLD related fibrosis. We performed three meta-analyses of the accuracy in detecting advanced fibrosis, including 11 studies. Due to a lack of sufficient accuracy data, no meta-analysis was performed for target condition significant fibrosis. FibroMeter VCTE showed the highest sensitivity and specificity in detecting advanced fibrosis. Interestingly, we observed better accuracy of FibroMeter V2G than FibroMeter NAFLD, which is specifically designed for NAFLD patients, in detecting advanced fibrosis. Yet, no statistically significant differences between the FibroMeter V2G and NAFLD versions were observed in our meta-regression analyses.

### 4.1. Published Literature

Currently, several non-invasive biomarkers for detecting different fibrosis levels are being used in the clinic and some of them, such as the Enhanced Liver Fibrosis score (ELF) and FibroTest, are recommended by clinical NAFLD guidelines [6,7]. These tests are complex panels consisting of different blood markers and clinical patient data such as age and BMI. Guillaume and colleagues compared the direct markers measured in ELF and FibroMeter V2G with two simple fibrosis scores, FIB-4 and NFS. The results of their study show that the tests with direct markers performed better than the more simple markers like FIB-4 and NFS in detecting advanced fibrosis in NAFLD patients [12]. This suggests that the inclusion of a direct marker for liver fibrosis in a non-invasive test may improve diagnostic accuracy. The results of the current meta-analyses are in line with this evidence, since we observed better accuracy for the FibroMeter V2G than for FibroMeter NAFLD. This is consistent with the findings of an earlier study performed by Boursier et al. [13]. They explained this difference by the inclusion of direct markers in the FibroMeter V2G, whereas FibroMeter NAFLD is calculated with only indirect markers of liver fibrosis. The direct markers of liver fibrosis included in FibroMeter V2G are hyaluronate and alpha2-macroglobulin. Hyaluronate levels rise during liver cell injury and fibrogenesis due to increased collagen turnover and reduced hepatic clearance, and alpha2-macroglobulin levels are also associated with hepatic collagen [35,36].

There is more evidence suggesting that a combination of routine blood-based variables and direct fibrogenesis markers can reach higher diagnostic accuracy in detecting advanced fibrosis [37,38]. Two studies reviewed various blood-based biomarkers and their diagnostic accuracy to recommend an algorithm to non-invasively assess patients with NAFLD and fibrosis. In their recommendation for clinical care, FibroMeter was one of the suggested tests to exclude or diagnose advanced fibrosis in patients with NAFLD at an intermediate risk category as determined by NFS and fatty liver index (FLI) [39]

The FibroScan-AST (FAST) score is another multi-component test, similar to FibroMeter VCTE, which includes AST blood marker and VCTE assessment. However, the FAST score only includes AST as blood marker and no direct fibrogenesis markers as included in FibroMeter VCTE. The performance of the FAST score in a derivation cohort showed an AUROC of 0.80 [40], which is lower when compared to the FibroMeter VCTE performance with an AUC of 0.94 in our study.

We performed an outlier analysis for both FibroMeter NAFLD and VCTE, because we observed remarkably low sensitivity reported for both FibroMeter tests in the study conducted by Loong et al. in a Chinese population [29]. This difference could not be explained by a difference in prevalence, because that was comparable to the other articles. However, a notable difference was that this study was one of the two studies (along with Yang et al.) conducted in a Chinese population. These two studies also reported the lowest mean BMI values, as reported in Table 1 [29,32]. The reported low sensitivities could also be due to limited performance of the used biometrics in FibroMeter in ethnicities other than Caucasian, since FibroMeter NAFLD and VCTE have been developed and validated mostly in Caucasian populations.

### 4.2. Strengths and Limitations

The included study groups were homogenous in age, BMI, and setting, partly because of our inclusion criteria. However, the studies included in our meta-analysis recruited their patients mostly from tertiary care centres, which could have led to higher prevalence of severe cases in our study population compared to primary and secondary care settings. The age in this systematic review ranges from 37 to 65 years. Note that in older patients non-invasive tests may be less accurate. We included two abstracts, which are not thoroughly peer-reviewed as full research articles, and this could be a limitation of the study. Two studies mentioned a conflict of interest. However, it is difficult to say what the impact of this conflict of interest is on our study results. The included studies reported a wide range of advanced fibrosis prevalence, ranging from 20% to 41%. This high prevalence should be considered when extrapolating these results to patient populations from primary and secondary care, with a lower prevalence of advanced fibrosis. Moreover, due to the lack of studies with head-to-head comparisons between different FibroMeter versions, direct comparison of FibroMeter versions was not possible. The limited number of studies available for meta-analysis impeded formal explorations of sources of heterogeneity. According to our QUADAS-2 tool assessment, the risk of methodological bias was low for most studies in the patient selection, reference standard, and flow and timing domains, with no applicability concerns in any of the studies, which strengthens the reliability of our results.

### 4.3. Implications and Future Perspective

There is a great need to detect advanced liver fibrosis in patients with NAFLD at an early stage to implement prevention and surveillance. The limitations of a liver biopsy increase the need for non-invasive tests to detect and stage liver fibrosis. In accordance, the EASL-EASD-EASO Clinical Practice Guidelines recommend FibroMeter as a non-invasive alternative to detect patients with a worse prognosis [7]. However, no clear recommendations for a specific FibroMeter version were made. In the current study we conducted an indirect comparison of different FibroMeter versions. Therefore, our first recommendation for future research would be to conduct head-to-head comparison studies of high methodological quality of the different FibroMeter versions in patients with NAFLD. These studies could contribute further to decisions about the preferred FibroMeter version in clinical care paths for different target conditions within the NAFLD spectrum. Such studies may reveal potential differences between FibroMeter V2G and FibroMeter VCTE, since in our study these FibroMeter tests had similar sensitivity estimates. Another interesting potential contribution from these studies would be the recommendation of a single cut-off value per FibroMeter, since we observed a number of different cut-off values used in the included studies. Calès and colleagues combined FibroMeter V2G and CirrhoMeter V2G into a score called Multi-FibroMeter. This score showed significant increases in the AUROC for cirrhosis compared to FibroMeter V2G, which leads to the conclusion that, in the recommended head-to-head comparison studies, Multi-FibroMeter should also have a role [41].

The current European guideline recommends a combination of tests to detect NAFLD severity, such as an ultrasound followed by a serum fibrosis marker in patients at risk of NAFLD [7]. Recently, Srivastava and colleagues introduced their care path, based on a sequential use of tests for evaluating patients with NAFLD. They reported that using an ultrasound assessment followed by FIB-4 and ELF could distinguish between patients with NAFLD who should be referred to second line health care and patients that could stay in the first line [42]. However, in clinical practice, ELF is not accessible for first line health care workers outside the United Kingdom. Interestingly, the recommended two-tiered approach of an ultrasound followed by a serum fibrosis marker is also captured by the FibroMeter VCTE, which combines FibroScan with FibroMeter V3G. In our study, FibroMeter VCTE had the highest average sensitivity and specificity when compared to the other FibroMeter versions in detecting advanced fibrosis in patients with NAFLD, although differences were not statistically significant. This leads us to the second recommendation for further research, to focus on clinical care paths with FibroMeter VCTE for patients with NAFLD fibrosis.

## 5. Conclusions

This systematic review and meta-analysis suggest that FibroMeter can have acceptable sensitivity and specificity in detecting advanced fibrosis in NAFLD patients, especially when combined with VCTE. However, statistical uncertainty remains, especially for sensitivity.

## Figures and Tables

**Figure 1 jcm-10-02910-f001:**
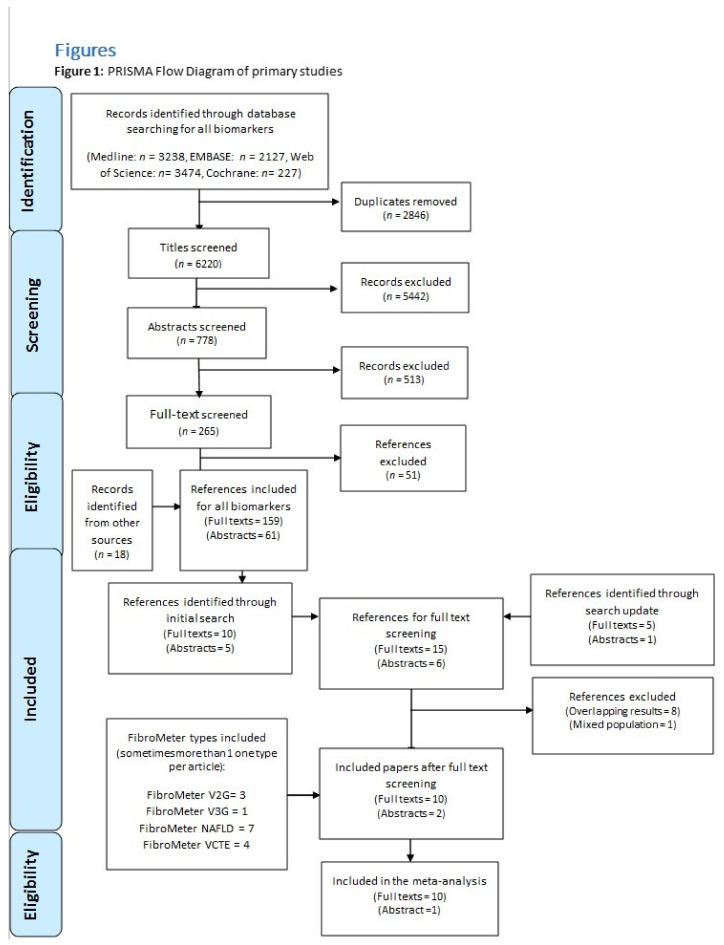
PRISMA Flow Diagram of primary studies.

**Table 1 jcm-10-02910-t001:** Characteristics of included studies.

	Study ID	Setting	Country	Population	*N* (Females)	Index Test (s)	Mean Age (years)	Mean BMI (kg/m^2^)	Target Condition/N (%)	Scoring System	ALT (U/L)	T2D	
1	Aykut 2014 [17]	Tertiary care centers	Turkey	Suspected NAFLD	88 (38)	FibroMeter NAFLD	46	30.3	Significant fibrosis F ≥ 2/44 (50%) Advanced fibrosis F ≥ 3/27 (31%) Cirrhosis F = 4/9 (10%)	NASH CRN	84	NR	
2	Boursier 2016 [13]	Tertiary care centers	France	Biopsy proven NAFLD	452 (181)	FibroMeter V2G FibroMeter NAFLD	56	31.1	Significant fibrosis F ≥ 2/NR Advanced fibrosis F ≥ 3/172 (38%)	NASH CRN	68	46.7%	
3	Boursier 2019 [26]	Tertiary care centers	France	Biopsy proven NAFLD	938 (389)	FibroMeter V2G FibroMeter VCTE	57	31.8	Significant fibrosis F ≥ 2/635 (68%) Advanced fibrosis F ≥ 3/383 (41%)	NASH CRN	56	51.1%	
4	Cales 2009 [27]	Tertiary care centers	France	Biopsy proven NAFLD	235 (60)	FibroMeter NAFLD	51	28.7	Significant fibrosis F ≥ 2/65 (28%) ‡ Advanced fibrosis F ≥ 3/44 (19%) ‡	METAVIR	NR	24.1%	
5	Dincses 2015 [28]	Tertiary care centers	Turkey	Biopsy proven NAFLD	52 (22)	FibroMeter VCTE	45	30.8	Significant fibrosis F ≥ 2/20 (38%) Advanced fibrosis F ≥ 3/10 (19%)	NASH CRN	89	36.5%	
6	Eddowes 2017 [33] †	NR	UK	Biopsy proven NAFLD	266(NR)	FibroMeter VCTE	NR §	NR	Advanced fibrosis F ≥ 3/106 (35%)	NR	NR	NR	
7	Loong 2017 [29]	Tertiary care centers	China	Suspected NAFLD	253 (96)	FibroMeter NAFLD FibroMeter VCTE	52	26.8	Significant fibrosis F ≥ 2/69 (32%) Advanced fibrosis F ≥ 3/43 (20%) Cirrhosis F = 4/20 (9%)	NASH CRN	58	54.9%	
8	Sanyal 2016 [34] †	Tertiary care centers	NR	Biopsy proven NAFLD	216 (96)	FibroMeter NAFLD	52	31.2	NASH with significant fibrosis (NAS ≥ 4; F ≥ 2)/95 (44%)	NASH CRN	62	36.6%	
9	Siddiqui 2016 [30]	Tertiary care centers	USA	Biopsy proven NAFLD	145 (92)	FibroMeter NAFLD	53	35.8	Any fibrosis F ≥ 1/103 (71%) Advanced fibrosis F ≥ 3/51 (35%)	NASH CRN	81	39.3%	
10	Staufer 2019 [31]	Tertiary care centers	Austria	Suspected NAFLD	186 (80)	FibroMeter V2G FibroMeter V3G	52	30.5	Significant fibrosis F ≥ 2/71 (54%) Advanced fibrosis F ≥ 3/49 (37%) NASH + Advanced fibrosis F ≥ 3/35 (27%) Cirrhosis F ≥ 4/20 (15%)	NASH CRN	55	30.0%	
11	Subasi 2015 [18]	Tertiary care centers	Turkey	Biopsy proven NAFLD	142 (67)	FibroMeter NAFLD	45	30.9	Advanced fibrosis F ≥ 3/30 (21%)	NASH CRN	91	16.9%	
12	Yang 2019 [32]	Tertiary care centers	China	Biopsy proven NAFLD	453 (186)	FibroMeter NAFLD	37	25.9	Significant fibrosis F ≥ 2/208 (46%) Advanced fibrosis F ≥ 3/126 (28%)	NASH CRN	135	30.2%	

† Abstracts; ‡ presented as reported as in the study of Cales 2009 in METAVIR classification, for analyses transformed in CRN criteria; § reported as 36–65 years (211 participants) and >65 years (55 participants); reported as median age instead of mean age. There is no overlap in the studies of Boursier et al. BMI = body mass index; T2D = type 2 diabetes mellitus prevalence; NR = not reported.

**Table 2 jcm-10-02910-t002:** Studies reporting AUC of target condition significant fibrosis (F ≥ 2) and their used cut-offs.

	Study ID	FibroMeter	Prevalence	Cut-off	AUC	95% CI AUC	Sensitivity	95% CI Sensitivity	Specificity	95% CI Specificity
1	Aykut 2014 [17]	FibroMeter NAFLD	0.50	0.62	0.62	0.06	0.39	0.25–0.54	0.86	0.72–0.94
2	Boursier 2016a [13]	FibroMeter V2G	NR	NR	0.79	0.02	NR	NR	NR	NR
2	Boursier 2016b [13]	FibroMeter NAFLD	NR	NR	0.76	0.02	NR	NR	NR	NR
3	Boursier 2019a [26]	FibroMeter V2G	0.68	0.77	0.75	0.02	NR	NR	NR	NR
3	Boursier 2019b [26]	FibroMeter VCTE	0.68	0.69	0.83	0.01	NR	NR	NR	NR
4	Dincses 2015 [28]	FibroMeter VCTE	0.38	0.38	0.82	0.06	0.70	0.46–0.87	0.88	0.70–0.96
5	Loong 2017a [29]	FibroMeter NAFLD	0.32	0.62	0.78	0.04	0.52	0.41–0.64	0.86	0.80–0.91
5	Loong 2017b [29]	FibroMeter VCTE	0.32	0.38	0.86	0.04	0.55	0.43–0.66	0.95	0.90–0.98
6	Staufer 2019 [31]	FibroMeter V2G	0.54	0.31	0.86	0.79–0.93	0.80	0.69–0.88	0.80	0.67–0.89
6	Staufer 2019 [31]	FibroMeter V3G	0.54	0.39	0.84	0.77–0.92	0.78	0.66–0.86	0.80	0.68–0.89

In total 8 studies mentioned significant fibrosis of which: 1 mentioned significant fibrosis with NASH (Sanyal 2016, not reported in this table) and 1 mentioned significant fibrosis according to METAVIR scoring system, which was calculated to advanced fibrosis in NASH CRN scoring system (Cales 2009, not reported in this table). AUC = area under the curve; 95% CI = 95% confidence interval; NR = not reported.

## Data Availability

The data that support the findings of this study are available from the corresponding author, A.-M.v.D., upon reasonable request. The protocol of the LITMUS systematic review is available in PROSPERO: CRD42018106821 https://www.crd.york.ac.uk/PROSPERO/display_record.php?RecordID=106821 (accessed on 14 June 2021).

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
