# Peer review of "Systematic Review with Meta-Analyses: Diagnostic Accuracy of FibroMeter Tests in Patients with Non-Alcoholic Fatty Liver Disease"

_jcm, 2021, doi:10.3390/jcm10132910_

Round 1
Reviewer 1 Report
In the current study, van Dijk et al. systematically review the literature on the diagnostic accuracy of the FibroMeter versions V2G, NAFLD and VCTE in detecting different levels of fibrosis in patients with NAFLD. They report that the different FibroMeter tests have acceptable sensitivity and specificity in detecting advanced fibrosis in NAFLD. FibroMeter VCTE was found to have the highest sensitivity and specificity, but differences were not statistically significant when compared to the other FibroMeter versions. Overall, this is a well-written manuscript addressing an important topic. Some issues deserve further attention as detailed below.
- The inclusion of conference abstracts, which are not thoroughly peer-reviewed as full research articles and may lack key information (e.g., in Ref. 34, fibrosis classification was not specified), in the systematic review is not acceptable.
- In the inclusion/exclusion criteria, have the authors considered the source of funding in each study? Potential conflicts of interest were found in a number of papers that were included in the current systematic review. This issue should be considered and addressed in the Methods, Results and Discussion.
- The mean age of the participants in the current systematic review varied from 37 to 65 yrs. Some non-invasive tests for assessment of liver fibrosis are known to have reduced accuracy in older patients (>65). This issue should be addressed.
- Supplementary Table 3 is not clear. Was F2 defined as “no/mild fibrosis’ in the ‘Present Study’?
- Please explain why additional sensitivity analyses were not performed for FibroMeter V2G (Figure 2).
- The Abstract could be improved: a) Abbreviation should be defined including those used for the different FibroMeter versions. b) The databases used for the literature search could be described. C) Please report in the Abstract that no statistically significant differences were found between the different FibroMeter versions.
- Fig. 3b (additional sensitivity analysis) is described in the text only after Fig. 4a. Please improve the flow of the text and figure order.
- The same references are numbered differently in the main text and supplementary data which is quite confusing.
- Some of the listed references have missing details including issue and pages (e.g., Ref. 29, 30, 32, 33 and others)
- What is the meaning of the numbers in the Keywords?
Reviewer 2 Report
The current manuscript aims to conduct a systemic review of publications that investigates the diagnostic performance of non-invasive FibroMeter tests in NAFLD patients with meta-analysis. NAFLD patients with liver fibrosis have higher morbidity and mortality. Currently, the gold standard of liver fibrosis is liver biopsy, which is invasive and highly depends on the degree of expertise of the observer. Several FibroMeter modalities, containing biometrics and blood parameters, have been studied for their functions to assist diagnosis of liver biopsy. In this article, the authors searched 3 databases for eligible studies for systemic review and meta-analysis. Of the 9066 studies, 12 were identified to meet the criteria of enrollment with a wide geographic range including multiple counties in Europe, the USA, and China. Due to the inability to rebuild the 2X2 contingency table for patients with significant fibrosis, the authors provided a systemic review of AUC levels and their corresponding cut-off values. For patients with advanced fibrosis, the authors performed a meta-analysis, showing FibroMeter VCTE with the best AUC (0.94), followed by FibroMeter V2G (0.83), and FibroMeter NAFLD (0.82). In summary, non-invasive FibroMeter tests have their diagnostic value in NAFLD patients in the tertiary center to assist liver biopsy for the diagnosis of liver fibrosis.
Although the study provides an important systemic review and meta-analysis of diagnostic value of FibroMeter tests in NAFLD patients, several concerns need to be addressed:
- In Table-1, it is mentioned that all the studies included in the review investigated patients in the tertiary care centers. This has the potential to introduce selection bias, as mentioned in “Strengths and limitations”. Thus, a better definition of the patient population that this study investigates should be described in the introduction section rather than discussing at the very end of the manuscript. This will better enable clarity for the readers.
- In Table-2, the definition of 95%CI of AUC is inconsistent, with the first 1-5 articles showing the standard error, while article 6 shows the true range of 95%CI. It is necessary to maintain a uniform format of 95%CI that will help reduce misinterpreting the message.
- In Supplementary Table 3, there is no term of “significant fibrosis”, as mentioned in the manuscript. It is necessary to match the definition of significant fibrosis(F≥2) with the stratification listed here.
- Throughout the manuscript, the authors describe AUC as diagnostic accuracy, which is not correct. Despite the common situation in which a higher accuracy comes with a higher AUC, they are completely two different terms. Accuracy = (correctly predicted class / total testing class) × 100%= (true positive + true negative)/total cases x 100%. It would be appropriate to use another term than “accuracy” to address the power of FibroMeter in liver fibrosis diagnosis represented by AUC. Can the authors comment on this and revise?
- A meta-analysis test should consist of a test for the statistical heterogeneity, for example, the chi-square test in Figure 2 of Passiglia et al, Sci Rep. 2018[PMID: 30190486], or Figure 3 of Yang et al, Sci Rep. 2021[PMID: 33510347]. However, all the forest plots of Supplementary figure 3-5 do not have a test for heterogeneity. It is therefore required to revise the figures to add the combined sensitivity/specificity with 95%CI plus the test result for heterogeneity at the bottom of each forest plot.
